# Osteopontin: A Promising Therapeutic Target in Cardiac Fibrosis

**DOI:** 10.3390/cells8121558

**Published:** 2019-12-03

**Authors:** Iman Abdelaziz Mohamed, Alain-Pierre Gadeau, Anwarul Hasan, Nabeel Abdulrahman, Fatima Mraiche

**Affiliations:** 1Visiting Scholar, Center of Excellence for Stem Cells and Regenerative Medicine (CESC), Zewail City of Science and Technology, 6th of October City, P.O. Box 12588 Giza Governorate, Egypt; iman.a.aziz@gmail.com; 2INSERM, Biology of Cardiovascular Disease, University of Bordeaux, U1034 Pessac, France; alain.gadeau@inserm.fr; 3Department of Mechanical and Industrial Engineering, College of Engineering, Qatar University, P.O. Box 2713 Doha, Qatar; ahasan@qu.edu.qa; 4Biomedical Research Center (BRC), Qatar University, P.O. Box 2713 Doha, Qatar; 5Translational Research Institute, Academic Health System, Hamad Medical Corporation, P.O. Box 3050 Doha, Qatar; kcnabeel87@gmail.com; 6Department of Pharmaceutical Sciences, College of Pharmacy, QU Health, Qatar University, P.O. Box 2713 Doha, Qatar

**Keywords:** osteopontin, inflammation, cardiac fibrosis, potential therapeutic target, biomarker

## Abstract

Osteopontin (OPN) is recognized for its significant roles in both physiological and pathological processes. Initially, OPN was recognized as a cytokine with pro-inflammatory actions. More recently, OPN has emerged as a matricellular protein of the extracellular matrix (ECM). OPN is also known to be a substrate for proteolytic cleavage by several proteases that form an integral part of the ECM. In the adult heart under physiological conditions, basal levels of OPN are expressed. Increased expression of OPN has been correlated with the progression of cardiac remodeling and fibrosis to heart failure and the severity of the condition. The intricate process by which OPN mediates its effects include the coordination of intracellular signals necessary for the differentiation of fibroblasts into myofibroblasts, promoting angiogenesis, wound healing, and tissue regeneration. In this review, we discuss the role of OPN in contributing to the development of cardiac fibrosis and its suitability as a therapeutic target.

## 1. Introduction 

Cardiovascular diseases are the number one cause of death globally with more people dying annually from cardiovascular diseases than from any other cause. [1,2]. Unlike most tissues, the heart is unable to repair itself because of the lack of sufficient cardiomyocyte proliferation. Wound healing plays a critical role in maintaining adequate heart function following cardiomyocyte death. This includes chronic extracellular matrix (ECM) deposition by myofibroblasts and further expansion of the scar [3]. Cardiac fibrosis is characterized by net accumulation of extracellular matrix proteins in the cardiac interstitium and contributes to both systolic and diastolic dysfunction in many cardiac pathophysiologic conditions [3]. It is a common theme in several types of heart diseases, including inherited cardiomyopathies, ischemic heart disease, diabetes and obesity, and ageing, and has been linked to morbidity and mortality [3]. During cardiac fibrosis, cardiac fibroblasts transform to a myofibroblast phenotype [3,4]. These myofibroblasts are responsible for the production of the extracellular matrix (ECM) and activation of several inflammatory pathways [5]. The early stages of this “healing” process promote the formation of a scar. The scar tissue is gradually substituted with new cells [6]. Failure to terminate the wound-healing program provokes a cascade of pathological changes that consequently result in cardiomyocyte hypertrophy, apoptosis, chamber dilatation, and ultimately, the development of congestive heart failure [3]. As a result, the interconversion of fibroblasts to myofibroblasts is prolonged. Alterations of the myocardial architecture of the injured heart contributes to impaired cardiac function and ventricular stiffness, leading to contractile dysfunction [4]. The accumulation of the ECM can also alter the mechano-electric coupling of cardiomyocytes, thereby amplifying the risk of arrhythmogenicity. This in turn exacerbates the progression towards heart failure and even sudden cardiac death [5]. Moreover, in dilated cardiomyopathy (DCM), elevated collagen synthesis and degradation have also been reported in the pathology of ECM fibrosis [7]. ECM fibrosis has been characterized by an overexpression of matrix metalloproteinases (MMPs) [7,8,9]. Although activated myofibroblasts are the main effector cells in the fibrotic heart, monocytes/macrophages, lymphocytes, mast cells, vascular cells, and cardiomyocytes may also contribute to the fibrotic response by secreting key fibrogenic mediators [5,8,9].

Regardless of the pathophysiologic injury leading to fibrotic remodeling of the ventricle, the networks of molecular signals involved are similar in various cardiac diseases [5,8]. Indeed, the relative contribution of each pathway is often dependent on the underlying cause of fibrotic remodeling [5]. Inflammatory cytokines and chemokines, reactive oxygen species, mast cell-derived proteases, endothelin-1, the renin/angiotensin/aldosterone system, matricellular proteins, and growth factors (such as transforming growth factor beta (TGF-β)) are implicated in cardiac fibrosis [8,9]. Inflammatory signals seem to be more important in reparative and ischemic fibrosis, while angiotensin/aldosterone axis and fibrogenic growth factors, such as TGF-β, appear to be involved in most fibrotic cardiac conditions regardless of the etiology [5]. Understanding the mechanisms responsible for the initiation and subsequent progression of cardiac fibrosis are crucial to identify effective anti-fibrotic treatment options. It has been demonstrated that cardiac injury promotes the activation of the renin–angiotensin–aldosterone system (RAAS), of which angiotensin II (Ang-II) appears to be the principal effector [4]. Ang II is heavily involved with the inflammatory response since it is activated and expressed by both macrophages as well as myofibroblasts [3]. In turn, this is thought to induce transforming growth factor β (TGF-β) signaling, which promotes the expression of genes that are characteristic of myofibroblast transdifferentiation, including α-smooth muscle actin, the extra domain-A fibronectin (ED-A FN), endothelin 1, connective tissue growth factor, and osteopontin (OPN), all which also serve as promoters of wound healing and fibrotic changes following cardiac injury [5,10]. ACE inhibition and AT1 blockade in patients with chronic heart failure or acute myocardial infarction has demonstrated to be beneficial, which in part maybe due to the inhibition of the angiotensin-induced fibrogenic actions. Aldosterone has also been demonstrated to induce fibrotic changes in the myocardium [5].In addition, the expression of the pro-inflammatory cytokines, such as TNF-α, interleukin 1 beta (IL-1β), and IL-6, are consistently induced in fibrotic hearts [5].Clearly, understanding the mechanisms that contribute to cardiac fibrosis provides further direction in identifying novel therapeutic interventions.

OPN plays an important role in a variety of cellular activities associated with inflammatory and fibrotic cascades, as well as wound healing [11,12]. It could act both as a matricellular protein (when bound to the matrix) and as a cytokine, when secreted in a soluble form [5]. In the unstressed heart, rat ventricular cardiomyocytes have been shown to express basal levels of OPN [13,14]. Interestingly, OPN expression has been demonstrated to be elevated upon the onset of cardiac hypertrophy (CH), suggesting its role as an effector for extracellular signaling, which induces myocyte growth [15]. In fibrotic lesions and healing wounds, both of which are characterized by myofibroblast differentiation, the expression of OPN is also markedly high [16]. Increased OPN levels have also been demonstrated both in the myocardium and plasma of patients with both DCM and hypertrophic cardiomyopathy (HCM), seemingly as a result of fibroblast activation [16]. OPN expression is also marked in heart failure of both ischemic and non-ischemic origin, as well as myocardial infarction (MI) [5,17]. In several experimental animal models, including MI, ischemia perfusion injury, and Ang-II or aldosterone-infusion, the downregulation of OPN was associated with a marked reduction in myocyte apoptosis and improved cardiac function [14,17]. Although elevated OPN activity has been detected in both myocytes and fibroblasts experiencing stress, there appears to be no indication of the long-term and lasting effects of blocking OPN signaling on the heart [15]. Therefore, a better understanding of the role of OPN in the pathogenesis of cardiac fibrosis, a common theme in various types of cardiovascular diseases, could hold the key to the identification of promising therapeutic targets for the treatment of patients with heart failure.

## 2. Osteopontin Structure

OPN was initially described in 1979 as a “secreted phosphoprotein” associated with malignant transformations in cancerous cell lines [18]. In 1985, it was first described as “sialoprotein” after its isolation from osteoblasts and osteoclasts [19]. A year later, in 1986, its cell binding sequence was identified and it was renamed OPN, meaning “bone-bridge” [19]. The expression of OPN was also detected in other cell types, including fibroblasts and macrophages [14], demonstrating its inflammatory role and interaction with the immune system [20]. OPN is an acidic-hydrophilic member of the matricellular protein family, a class of non-structural ECM proteins [17]. Alternative splicing of OPN results in three isoforms, OPN-a (the full-length isoform), OPN-b (which lacks exon 5), and OPN-c (which lacks exon 4) [21]. The full-length human OPN protein is made up of 314 amino acid residues with a predicted molecular mass of around 32 kDa [22], while the OPN-b and OPN-c isoforms are made up of 300 and 287 amino acids, respectively [23]. The functional significance of deleting this amino acid segment is unknown. The OPN coding sequence seems to be highly conserved amongst species [14]. The apparent molecular weight of OPN can range from 45–75 kDa on gel electrophoresis due to extensive post-translational modification (PTM) caused by phosphorylation, glycosylation, and sulfation [14,19,24]. This has also been attributed to the high density of negative charges throughout the protein from the predominance of acidic amino acids, as well as the multitude of serine phosphorylation sites [22]. The significance of PTM is to facilitate the activation of OPN [19,20]. It has also been suggested that PTM leads to tissue-specific isoforms of OPN [25]. The diverse biological functions of OPN can be attributable due to its structural features, PTMs, interaction with multiple receptors, and different isoforms [5,17]. Alternative translation of OPN generates two isoforms: A cell-secreted full length OPN (*sOPN*) and a shortened protein lacking the N-terminal signal sequence of sOPN, intracellular (*iOPN*), found in the cytoplasm and nucleus [26]. Studies investigating the involvement of OPN in cardiovascular diseases primarily focus on total OPN expression and do not take into account the expression levels of any specific OPN isoform [21].

OPN carries various functional domains, which allow for receptor binding for the promotion of several biological functions. The N-terminal contains the arginine-glycine-aspartate (RGD) cell binding sequence (Figure 1) [22]. The N-terminal also carries a non-RGD serine-valine-valine-tyrosine-glutamate-leucine-arginine (*SVVYGLR*) domain between the RGD cell binding sequence and the thrombin cleavage site. This domain allows for the interaction between OPN and integrin receptors α4β1 and α9β1, respectively [22]. OPN also contains two recognized heparin bindings sites as well as a calcium (Ca^2+^) binding site near the C-terminal of the protein (Figure 1) [24]. Additionally, transglutamination allows OPN to be cross-linked to itself or other proteins by using two of its highly conserved glutamine residues. The process of transglutamination increases OPN’s collagen-binding abilities [20].

Nuclear magnetic resonance studies have indicated that OPN is a biologically active protein that typically lacks a defined tertiary, and often secondary, structure [27]. Recent computer algorithms were able to predict the existence of some secondary structures composed of short α-helices and β-strand structures [25]. These abnormal conformities suggest the possible interactions of the C-terminal domain of OPN with the central RGD-SVVYGLR integrin-binding domain [25].

## 3. Physiological Function of OPN

OPN has emerged as a key ubiquitous phosphoprotein involved in numerous biological functions, including biomineralization, ECM and tissue remodeling and repair, immunity and the inflammatory response, cell death, and survival (Figure 2) [24,26,28]. OPN’s conformational flexibility mediates its interaction with a multitude of ligands across a dynamic array of cellular surfaces [27]. The interaction of the *N*-domain with αvβ1, αvβ3, and αvβ5 integrin receptors has been shown to mediate cell migration [20,28]. In rat aortic smooth muscle cells, OPN mediates platelet-derived growth factor migration from the media to the intima of arteries [29]. It is strongly expressed during injury-induced vessel calcification and is vital for inhibiting this vascular calcification in smooth muscle cells in mice [25]. Interestingly, PTM of OPN can impact its biological functions [17], where the un-phosphorylated or enzymatically dephosphorylated OPN appears to have no effect in mice [14] and the native phosphorylated form of OPN is able to inhibit calcification of human smooth muscle cells in vitro [17]. Moreover, OPN promotes macrophage and endothelial cell migration and proliferation, thus contributing to vasculature remodeling [30].

OPN is also an important regulator of apoptosis [28]. OPN’s pro-survival properties have also been attributed to the binding of phosphoprotein with the αvβ3 integrin receptor, in addition to activation of downstream nuclear factor-kappa β (NFkB) signaling [31]. OPN is also thought to interact with hyaluronic acid receptor 44 (CD44) via the Ca^2+^ binding site [24], along with N-terminal, thereby allowing OPN to regulate CD44 surface expression [19,22]. The anti-apoptotic actions of OPN have also been attributed to its interaction with the CD44 receptor and activation of the downstream phosphoinositide 3 (PI-3)-kinase/protein kinase B (Akt) cascade [17].

OPN appears to play an important role in wound healing due to its effects on cell viability, as well as modulating the secretion of MMPs, and the proliferation and differentiation of fibroblasts [28]. At the wound site, OPN also serves as a chemotactic molecule to promote macrophage infiltration and recruit inflammatory cells, as well as acting as an adhesive protein to retain cells at the site of injury [30]. OPN’s role in promoting wound healing has also been attributed to its ability to bind directly to collagen type I and interact with collagen types II, III, IV, and V, as evident by the alterations in collagen fiber diameter observed in OPN-null mice enduring incisional wounds [30]. OPN is an important angiogenic factor, could act as a chemo-attractant, and is shown to promote vascular cell adhesion and spreading in vitro [22]. OPN’s ability to promote angiogenesis has been attributed to its action directly through PI3K/AKT- and extracellular signal-regulated kinase (ERK)-mediated pathways, which enhances the expression of vascular endothelial growth factor (VEGF), and vascular permeability factor (VPF), cytokines that have been shown to induce angiogenesis [21].

In bones, OPN is one of the more abundant non-collagenous proteins whose phosphorylation provides a stabilizing effect on the bond between calcium and hydroxyapatite, thus serving as an attachment protein linking cells to the bone mineral [25]. It also facilitates the attachment of osteoclasts to the bone matrix via an interaction with CD44 and cell surface αvβ3 integrin receptor. OPN has also been shown to act through the Ca^2+^ -NFAT (nuclear factor of activated T cells) pathway to promote osteoclast survival and increase bone resorption [22]. iOPN also plays a role in osteoclast motility, fusion, and resorption, and could be responsible for some of the observed effects of OPN deficiency, including bone loss due to reduced mechanical stress [25]. In the immune system, OPN facilitates chemotaxis, leading to the migration of macrophages and dendritic cells to sites of inflammation [28]. It has also been shown that OPN can modulate the immune response by macrophage recruitment and differentiation, enhanced expression of Th1 cytokines and matrix-degrading enzymes, as well as phagocytosis in mice [30]. OPN’s role in the inflammatory process has also been attributed to the binding of phosphoprotein with the integrin receptor, leading to NFkB-mediated expression of inflammatory cytokines [22].

As a matricellular protein, OPN appears to function as a modulator of cell–ECM interactions, often achieved by binding to growth factors, structural matrix proteins, proteases, and cell-surface receptors, rendering them essential components of the ECM environment [24]. One of the features of OPN’s structure that has a huge impact on its biological function is the presence of a thrombin-cleavage site, which allows the protein to be cleaved into two fragments: The N-terminal (amino) fragment carrying both RGD and SVVYGLR domains, and a C-terminal (carboxyl) fragment, which contains the Ca^2+^ and 2 heparin binding sites [19]. It appears that the N-terminal fragment possesses adhesion motifs that may render it even more biologically active than the full-length OPN protein whereas the properties of the C-terminal fragment of OPN are less well characterized [32]. OPN is also susceptible to cleavage by different proteolytic MMPs (MMP 2,3,7,9), which form an integral part of the ECM [32]. Each MMP yields a unique cleavage profile, with only few overlapping cleavage sites giving peptides that can exert distinct biological functions. Data from carotid-plaque samples of hypertensive patients have also recently suggested that the cleavage of OPN further enhances the ability of the RGD binding site to interact with the CD44 as well as αvβ1, αvβ3, and αvβ5 integrin receptors, thus magnifying the cell adherence capabilities of OPN [32], and rendering OPN an integral component for the homeostasis of the ECM environment.

## 4. Role of OPN in Heart Disease

For the heart to continue to function in a state of equilibrium, very intricate crosstalk between the different types of cells residing in the heart, as well as their interaction with the ECM, is orchestrated [33]. While cardiomyocytes constitute approximately two thirds of the myocardial tissue volume, cardiac fibroblasts actually have the highest cell population in the myocardium (accounting for approximately two thirds of the overall cells) [9]. OPN expression in the adult heart is detected at basal levels, as observed by northern blot analysis of fresh primary isolates of adult rat ventricular myocytes [14]. Fibroblasts and microvascular endothelial cells have also been shown to express low basal levels of OPN under physiological conditions [17]. The lack of OPN in experimental models of MI injury in rats showed signs of reduced repair as evident by reduced collagen synthesis and deposition in both infarct and non-infarct regions, resulting in ventricular dilation [34]. On the other hand, there appears to be a strong correlation between OPN levels and the severity of several cardiac pathologies [19]. Rubis et al. recently demonstrated that elevated levels of OPN were found to be related to more severe cardiovascular outcomes in an observational, prospective cohort study of DCM [7]. Moreover, human myocytes isolated from explanted hearts with either idiopathic or ischemic cardiomyopathy presenting with extensive fibrosis and CH demonstrated substantial immunoreactivity for OPN [14]. In fact, elevated expression of OPN has also been shown to be associated with the development of heart failure [35]. Thus, OPN emerges as a double-edged sword during cardiac adaptation and disease, where initially, it is heavily involved with cardiac repair following ischemic injury; on the other hand, OPN expression appears to be markedly increased in the heart under different pathological states.

Enhanced expression of OPN appears to play an important role in the remodeling of the heart post-MI, as well as in left ventricular hypertrophy [36,37,38,39]. One putative route through which the OPN could be mediating its hypertrophic effects is through the activation of pro-hypertrophic kinases and pathways. Interestingly, levels of phosphorylated Akt and glycogen synthase kinase-3β (GSK-3β) were significantly higher in mice subjected to aortic banding that develop CH [37]. Phosphorylation of GSK-3β enhances the translocation of NFAT and GATA-4 transcription factors to the nucleus, inducing hypertrophic gene expression [13,22]. The calcineurin (CaN)/nuclear factor of activated T-cells (NFAT) pathway has also been highlighted as a possible pathway for OPN to mediate CH [13,40]. This was emphasized in a study in which the activation of NFATc1 and GATA-3 (both pro-hypertrophic factors) were significantly reduced in OPN-deficient mice [41]. OPN has also recently been shown to mediate the hypertrophic response induced by cardiomyocyte expression of the activated form of Na^+^/H^+^ exchanger isoform 1 (NHE1) [13,39,40,42]. Transgenic mice expressing active, cardiac-specific NHE1 showed alterations in gene expression that led to both CH and an upregulation of OPN as well as its related molecules, including fibronectin, CD44, integrin receptors, MMPs, and signaling kinases [42]. Interestingly, transfection of cardiomyocytes with siRNA-OPN significantly abolished the NHE1-induced hypertrophic response, as well as significantly reducing the activity of NHE1 in cardiomyocytes expressing active NHE1 [13]. Moreover, in the double transgenic mice expressing active NHE1 in which OPN was knocked out, cardiac hypertrophic parameters, including increased collagen deposition, upregulation of CD44, and phosphorylation of p90 ribosomal s6 kinase (RSK), were significantly reduced when compared to NHE1 transgenic mice [39]. In another model where CH was induced by transverse aortic constriction (TAC) in serum- and glucocorticoid-inducible kinase (SGK1) SGK1^+/+^ mice, a marked increase in both OPN and NHE1 expression/activity was shown [43]. These findings further implicate OPN in mediating the hypertrophic effects observed in different models of CH.

OPN also appears to play a role in several cardiomyopathies, including diabetic cardiomyopathy, in vivo [44,45] DCM both in mice [46] and in humans [7] and in patients with HCM [26]. The implication of OPN in idiopathic cardiomyopathies was also observed in hearts explanted at the time of cardiac transplantation, revealing an upregulation of OPN in cardiomyocytes [47]. Although the exact mechanism through which OPN mediates diabetic cardiomyopathy is not fully understood, Nilsson-Berglund et al. recently identified two NFATc3 responsive sequences in the OPN promoter driving its expression in diabetic mice [45]. Conversely, OPN expression was significantly reduced in arteries from the diabetic NFAT knockout (KO) mice [45]. In another study, Subramanian et al. suggested that activation of protein kinase C (PKC)-βII could be mediating the upregulation of OPN in a mouse model of diabetic cardiomyopathy [44]. Interestingly, Soetikno et al. revealed that curcumin attenuated diabetes-induced cardiomyopathy in mice through the reduction of OPN expression and PKC-βII activity [48]. Collectively, these findings have important clinical implications, as they indicate that the inhibition of selected pathways or ligands involved in the OPN pathway may be of therapeutic benefit for the prevention of further cardiac damage and dysfunction.

OPN may also play an important part in the inflammatory process of atherosclerosis [22]. It has been suggested that OPN plays a role in the migration of endothelial cells, proliferation, and/or differentiation of vascular smooth muscle cells (VSMCs) along with plaque progression and dystrophic calcification [19]. OPN also has the ability to promote macrophage and VSMC accumulation, thus contributing to vasculature remodeling both in vivo and in humans [30]. An early study has demonstrated that VPF/VEGF activation in human dermal microvascular endothelial cells increases the expression of αvβ3 and its ligand OPN on the cell surface, which contributes to the migration of endothelial cells and macrophages [19]. OPN’s migration-promoting activity is further enhanced by thrombin cleavage of OPN as observed in carotid plaques from hypertensive patients [32]. Moreover, Lee et al. demonstrated that the human OPN isoforms OPN-a, OPN-b, or OPN-c, which are not expressed in rodents, may represent a novel therapeutic target to improve neovascualrization and preserve tissue function in patients with obstructive artery diseases [24]. [OPNa and OPNc significantly improved limb perfusion in OPN^−/−^ mice who underwent hindlimb ischemia surgery while OPN-a and OPN-c treated animals exhibited significant increases in arteriogenesis [24]. These findings suggest that the apparent role of OPN in both cardiac and vascular injuries make it an ideal therapeutic target to prevent irreversible damage to the myocardium.

## 5. Role of OPN in Cardiac Fibrosis

Injury to the heart can result in the amplification of resident cardiac fibroblasts, the transformation of endothelial/epithelial cells to fibroblasts, and even the recruitment of hematopoietic cells to the site of injury and their transformation into cardiac fibroblasts and myofibroblasts [49]. This cascade of complex cellular and molecular interactions determines successful recovery or inadequate repair of the damaged tissue in the heart. OPN is essential for the differentiation of fibroblasts into myofibroblasts post-MI injury, as documented and demonstrated by several studies [6,14,16,22,46,47,50,51,52,53,54,55]. These activated fibroblasts and myofibroblasts become the source of ECM proteins necessary for scar formation and healing of infarcted regions of the heart [6]. However, failure to terminate this wound-healing process results in persistent activation of fibroblasts and excessive cardiac remodeling associated with cardiomyocyte death, myocardial degeneration, impaired ventricular relaxation, and overall cardiac decompensation [8,10,15,49]. This cascade of events has been suggested to further increase the release of OPN from cardiomyocytes, thereby increasing OPN expression in infarct as well as non-infarct regions of the heart post-MI [56].

Activation of the renin-angiotensin-aldosterone system (RAAS), and consequent release of Ang-II post-MI has been shown to induce the expression of OPN in both neonatal and adult rat cardiac fibroblasts (NRCFs and ARCFs) [14]. Ang-II-induced activation of OPN has been suggested to occur in part through the ERK1/2 and c-Jun N-terminal kinases (JNK) pathways in ARCFs [57]. The production of reactive oxygen species appears to be a common mechanism involved in the increased expression of OPN from ARCFs post-MI [22]. Indeed, Ang-II induced significant phosphorylation of p42/44 mitogen-activated protein kinase (MAPK) and increased superoxide production, effects that were blocked by an NADPH oxidase inhibitor [14]. In addition, the inhibition of JNKs in ARCFs, combined with inhibition of ERK1/2, was able to almost completely inhibit Ang-II-mediated upregulation of OPN [57]. Taken together, these studies highlight some of the potential signaling pathways that mediate the Ang-II-induced increase of OPN expression in interstitial cells and the heart post-MI (Figure 3).

Many studies have demonstrated the efficacy of partial or complete OPN inhibition in a variety of preclinical models of diseases, including CH [15,37,39], coronary artery disease [24,28,58,59], DCM [46], diabetic cardiomyopathy [48], heart failure [52], and cardiac fibrosis [13,15,38,39,60]. The earliest study by Matsui et al. demonstrated that the Ang-II-induced elevation in blood pressure, collagen deposition, and development of cardiac fibrosis was significantly reduced in OPN^−/−^ mice compared to wild type (WT) mice [38]. However, the reduction of cardiac fibrosis led to an impairment of cardiac systolic function and subsequent left ventricular dilation in the OPN^−/−^ mice [38], suggesting that completely knocking out OPN in the heart can actually decrease the intrinsic repair mechanism following Ang-II infusion, resulting in side-to-side slippage of cardiomyocytes and the impairment observed. This is in line with previous results, which demonstrated that OPN deficiency in a murine model of MI caused exaggeration of left ventricular dilation and reduction of collagen deposition in comparison with WT mice [59]. These findings raise the question of whether the use of more specific inhibitors or deletion of specific OPN isoforms in the heart post-MI could be more beneficial for regressing cardiac fibrosis.

Using the streptozotocin-induced diabetic cardiomyopathy mouse model, Subramanian et al. demonstrated that the increase in fibrosis was significantly lower in OPN^-/-^ versus WT mice, which was also associated with lower TGF-β levels [48]. These findings suggest that the lack of OPN can actually be beneficial in reducing myocardial fibrosis in the diabetic heart. In another study where the genetic model of heart failure was examined (double desmin^−/−^OPN^−/−^ mice), the deficiency in OPN reduced fibrosis and improved the systolic properties of desmin^−/−^ myocardium [52]. Interestingly, in the NHE1-induced CH model of mice, knocking out OPN was sufficient to significantly reduce collagen deposition, upregulation of CD44, and phosphorylation of RSK, features that are characteristic of cardiac fibrosis [39]. Taken together, these findings suggest that benefits from knocking out OPN could be possible in specific settings of cardiac dysfunction, including diabetes, CH, and heart failure.

Recent genome-editing tools have emerged as powerful methods, which enable the generation of genetically modified cells and organisms necessary to elucidate gene function and mechanisms of human disease. Such tools include the use of RNA interference, specifically exogenously derived small interfering RNA (siRNA), and clustered regularly interspersed short palindromic repeats/CRISPR-associated protein 9 (CRISPR/Cas9) gene-editing technologies [61]. More specific means of OPN inhibition have also received special attention recently, including the use of siRNA technology. Mohamed et al. demonstrated that transfection of cardiomyocytes with siRNA-OPN significantly improved the NHE1-induced hypertrophic response in cardiomyocytes [13]. More specifically, inhibition of OPN using siRNA was able to significantly reduce cell surface area, total protein content, atrial natriuretic peptide (ANP) mRNA expression as well as GATA4 phosphorylation, compared to cardiomyocytes expressing active NHE1 alone [13]. The potential of silencing OPN as a means to revert CH is rather promising. However, it remains to be determined whether the use of siRNA against OPN would be beneficial in attenuating fibrosis in a post-MI model, without compromising cardiac function and left ventricular dilatation, as was previously demonstrated.

β-adrenergic receptor (AR) blockers have demonstrated significant survival benefits and have become standard therapy for adults battling ischemic heart disease. Current appreciation of β-receptor signaling involves activating pathways by which these receptors crosstalk with other signaling pathways; making the β1-AR the “cardiotoxic subtype” whereas the β2-AR appears to be the “cardioprotective” one [60]. While the more well-known β1-AR mediates the positive inotropic and chronotropic actions of the sympathetic nervous system, β2-AR has been shown to exert several cardioprotective effects in the heart post-MI, such as inhibition of fibrosis as well as inflammation and apoptosis [5]. As a Gs protein-coupled receptor (GPCR), β2-AR antifibrotic effects have been attributed to the stimulation and generation of cyclic 3′,5′-adenosine monophosphate (cAMP), which activates both protein kinase A (PKA) and Epac, an exchange protein directly activated by cAMP known to inhibit cardiac fibroblast activation and fibrosis [60]. Pollard et al. sought to delineate the role of OPN on β-AR-regulated cardiomyoblast fibrosis, using the H9c2 rat cardiac myoblast cell line [60]. Interestingly, the study demonstrated that CRISPR-mediated OPN deletion enriched the release of cAMP, leading to the upregulation of Epac protein levels, and abrogating TGF-β-induced fibrosis, in response to β2-AR activation in H9c2 cells [60]. This data further supports the notion that specific OPN-blockade can significantly boost the therapeutic efficacy of cardioprotective pathways, such as the β2-AR, against cardiac fibrosis.

The use of cell-permeable OPN function-blocking antibodies have revealed that not only is OPN an important contributor to myocardial fibrosis, but it could possibly mitigate remodeling of the heart following injury [15,56]. This was demonstrated in a recent study, in which the genetic model of heart failure (cardiac-specific integrin-linked kinase (ILK) knockout) mice were injected intraperitoneally with a neutralizing anti-mouse OPN polyclonal goat immunoglobulin G [56]. Upon preforming echocardiography, Dai et al. revealed that the OPN-specific antibody could improve the functional decline in cardiac-specific ILK knockout mice when compared to the control mice [56]. However, blocking OPN appeared to only mitigate rather than fully rescue the functional decline in the mice, suggesting that OPN treatment for a transient period of time alone may be insufficient to reverse cardiac dysfunction and fibrosis. In a later study, Li et al. demonstrated that blocking OPN at the time of surgeries with an aptamer (small structured single-stranded nucleic acid ligands, which have emerged as alternatives to antibodies) and at two months post-surgeries, in the mouse model of pressure overload, prevented cardiomyocyte hypertrophy and cardiac fibrosis, reduced expression of ECM components (FN1, lumican, and collagen type III α1 chain) and re-expression of the fetal genes, prevented cardiac dysfunction and blocked OPN downstream signaling (PI3K/Akt phosphorylation), and reversed cardiac dysfunction, fibrosis, and hypertrophy [15]. Taken together, these studies demonstrate that blocking and/or powering down cardiac OPN signaling in a time-specific manner may be more effective at reversing CH and fibrosis, thus improving cardiac function.

## 6. OPN as A Biomarker

Although several studies have demonstrated the efficacy of OPN inhibition in a variety of preclinical models of cardiovascular disease, clinical applications have not yet been demonstrated, since the detailed and long-term effects of blocking OPN signaling on the heart remain poorly understood. That being said, OPN is not only considered a pivotal protein controlling myocardial fibrosis, it is becoming increasingly popular as a biomarker associated with the prognosis of cardiovascular disease [7,28]. This has been demonstrated in a study by Dimitrow et al. in which a statistically significant decrease in OPN levels was observed in 33 patients battling heart disease following atorvastatin treatment for four weeks [28]. Correlations between OPN levels and diabetic vascular complications and all-cause mortality in patients with type 1 diabetes was also demonstrated in the FinnDiane study, in which serum OPN levels were significantly higher at baseline in patients who experienced a cardiovascular disease event compared to those who did not [62]. Collectively, these data demonstrate the potential of using OPN levels to detect the existence and severity of coronary artery disease and its complications.

OPN expression has also been correlated with the onset of heart failure in both animal models [12] and clinical settings [51,63]. Trueblood et al. demonstrated in a rat model that OPN levels were increased following the transition from left ventricular hypertrophy to cardiac decompensation [59]. The upregulation of OPN mRNA was suggested to originate from infiltrating macrophages and fibroblasts in the interstitium, suggesting that the role of OPN during heart failure could be attributed to its pro-inflammatory effects [14]. Clinically, during the early stages of acute ST-elevation MI and in patients who had undergone reperfusion within 12 h of the onset of anterior-wall acute MI, plasma levels of OPN have been shown to be elevated [64,65]. Additionally, in the subset of patients who underwent successful reperfusion following anterior-wall acute MI, reduction in OPN plasma levels was significantly correlated with the left ventricular end-systolic and diastolic volume index as well as ejection fraction [65]. Moreover, in patients undergoing cardiac resynchronization therapy (CRT), an effective treatment for heart failure in patients with ventricular dyssynchrony, plasma OPN levels were significantly lower in responders to CRT compared to non-responders [66]. Furthermore, in a recent study where the role of OPN in hypertension-induced heart failure was investigated, plasma levels of OPN appeared to be significantly higher in the hearts of patients with heart failure compared to control hearts [51]. Collectively, these observations suggest that elevated plasma levels of OPN could serve as a biomarker of the occurrence of an MI incident, as well as an indicator of the severity of heart failure and an indicator of the response to heart failure therapies.

## 7. Conclusions

Cardiac fibrosis is characterized by the net accumulation of extracellular matrix in the myocardium and is an integral component of most cardiac pathologic conditions. In our review article, we highlighted the relevance of OPN in contributing to the intercellular signals required to integrate myofibroblast proliferation, migration, and ECM deposition. In addition, we discussed the significance of OPN activity in both myocytes and fibroblasts experiencing stress, and that elevated plasma levels of OPN could serve as a biomarker of the occurrence of an MI incident, as well as an indicator of the severity of heart failure and the response to heart failure therapies. Furthermore, inhibition of selected pathways or ligands involved in the OPN pathway may be of therapeutic benefit for the prevention of further cardiac damage and dysfunction. These findings suggest the pivotal role of OPN in cardiac fibrosis and remodeling and that a better understanding of the role of OPN in the pathogenesis of cardiac fibrosis could hold the key to the identification of promising targets for the treatment of patients with heart failure. Whether the use of more specific inhibition or deletion of OPN in the heart post-MI could be more beneficial for cardiac fibrosis remains unclear.

## Figures and Tables

**Figure 1 cells-08-01558-f001:**
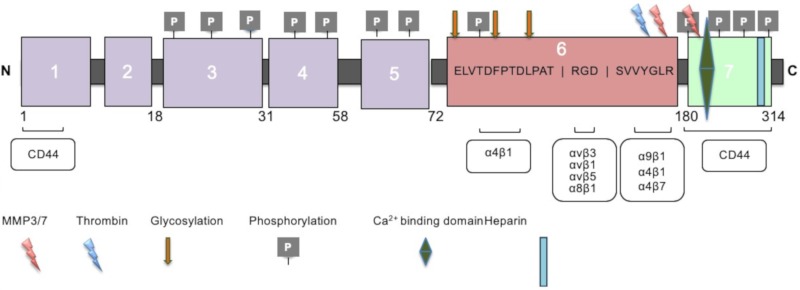
Structural features of the osteopontin (OPN) full-length secreted isoform with six translated exons (314 amino acids). OPN isoforms include the following functional domains: Arginine-glycine-aspartic acid (RGD) domain (159–161), serine-valine-valine-tyrosine-glutamate-leucine-arginine (SVVYGLR) domain (162–168), thrombin cleavage site, calcium binding domain (216–228), and heparin binding domain. Integrin binding occurs at RGD and SVVYGLR sequences. CD44 variant receptor binding occurs near the C-terminus and N-terminus. Phosphorylation sites exist across the precursor-mRNA (pre-mRNA). All OPN isoforms have thrombin and matrix metalloproteinase (MMP) cleavage sites.

**Figure 2 cells-08-01558-f002:**
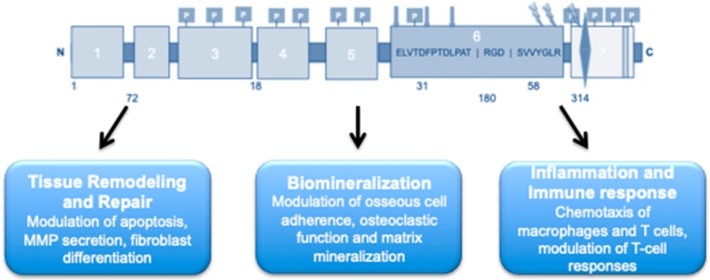
As a multifunctional protein, OPN plays a crucial role in several biological processes, including inflammation and the immune response (as a chemical attractant for macrophages and T cells, and a modulator of T-cell responses), biomineralization (by modulating the adherence of osseous cells, and modulation of both osteoclastic function and matrix mineralization), and tissue remodeling and repair (by modulating apoptosis, and secretion of metalloproteinase (MMPs) as well the regeneration and differentiation of fibroblasts and myofibroblasts).

**Figure 3 cells-08-01558-f003:**
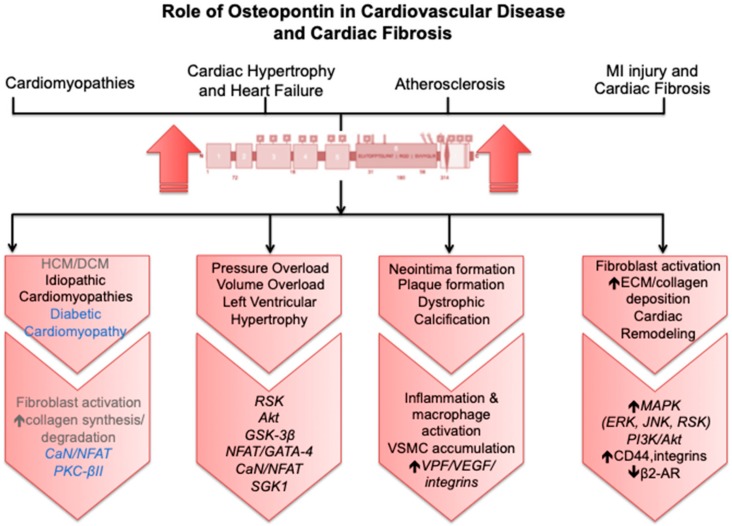
Role of osteopontin in cardiovascular disease and cardiac fibrosis, dilated cardiomyopathy (DCM), and hypertrophic cardiomyopathy (HCM), seemingly as a result of fibroblast activation. Elevated collagen synthesis and degradation have also been reported in the pathology of extracellular matrix (ECM) fibrosis. The calcineurin-NFAT pathway and activation of protein kinase C-βII could be mediating the upregulation of OPN in diabetic cardiomyopathy. In several models of cardiac hypertrophy and heart failure, OPN has been shown to induce the activation of several pro-hypertrophic kinases and pathways, including: p90 ribosomal s6 kinase, Akt, glycogen synthase kinase-3β, NFAT/GATA-4, calcineurin-NFAT, and serum- and glucocorticoid-inducible kinase. OPN has also been implicated in atherosclerosis and the process of neointima and plaque formation and dystrophic calcification by orchestrating the immune response and vascular smooth muscle cell migration. While upregulation of OPN in the heart following injury has the potential to protect against left ventricular dilation, it can also increase cardiac fibrosis and induce pathological remodeling through excessive myofibroblast differentiation, increased extracellular matrix, collagen synthesis, and deposition. This appears to involve the activation of downstream signaling pathways, including the mitogen-activated protein kinase (MAPK), extracellular signal-regulated kinase (ERK), c-Jun N-terminal kinases (JNK), and phosphorylation of RSK; as well as the PI3K/Akt pathway. The interaction between OPN, CD44, and integrins has also been suggested to further promote tissue fibrosis and decrease the cardioprotective effects of the β2-adrenergic receptor (β2AR).

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
