# Peer review of "Osteopontin: A Promising Therapeutic Target in Cardiac Fibrosis"

_cells, 2019, doi:10.3390/cells8121558_

Round 1
Reviewer 1 Report
All abbreviations should be spelled out, for example, OPN or ED-A. Following sentense (Lines 216 to 220) could not be understood well.
OPN has recently been shown to mediate the hypertrophic response of the activated form of the ubiquitously expressed membrane protein Na+/H+ exchanger isoform 1 (NHE1)-induced OPN has also recently been shown to mediate the hypertrophic response induced by cardiomyocyte expression of the activated form of Na+/H+ exchanger isoform 1 (NHE1)
(13, 36, 39, 41).
Author Response
We thank the reviewer for their valuable feedback. Please find below our detailed response to the comments. Please note corrections have been included in both the edited and the final versions of the submitted manuscript.
Reviewer #1:
All abbreviations should be spelled out, for example, OPN or ED-A.
Response:
Thank you. We have included the definition of all abbreviations.
Following sentences (Lines 216 to 220) could not be understood well.
Response:
Thank you. We have revised the sentences
Reviewer 2 Report
In the review „Osteopontin: a Promising Therapeutic Target in Cardiac Fibrosis”, the authors describe osteopontin (OPN) structure than its functions in physiological as well as in pathological context. They highlighted OPN implication in heart diseases and especially in cardiac fibrosis before offering an opening about OPN inhibition.
In general, this review is well documented and the topic is a great interest in the research community. Unfortunately, there are some structural problems in the manuscript:
In the part “osteopontin structure”, the OPN isoform notion is found line 109 when the explanation is in lines 117 to 121. In the same way, some references appear in the wrong order: reference 59 before the references 57 and 58 or reference 63 before reference 62. I would also be happy to see the relevant information from the references 61 and 65 which seems interesting for the manuscript but do not appear in the text.The introduction part is in my point of view well done despite an overusing of acronyms which are not evocated later in the text (for instance: I/R line 72, NMR line 112…)
The “osteopontin structure” part presents some imprecise information:
Line 80: OPN was first described in malignant cells in 1979, in normal tissues in 1985 and renamed OPN in 1986. All references and details can be found here: Lamort AS, Giopanou I, Psallidas I, Stathopoulos GT. Osteopontin as a Link between Inflammation and Cancer: The Thorax in the Spotlight. Cells. 2019 Aug 2;8(8) The OPN with 314 aa (line 84) corresponds only to isoform OPN-a, the isoforms OPN-b and –c are respectively 300 and 287 aa. Some details can be found here: Chae, S.; Jun, H.-O.; Lee, E.G.; Yang, S.-J.; Lee, D.C.; Jung, J.K.; Park, K.C.; Yeom, Y.I.; Kim, K.-W. Osteopontin splice variants differentially modulate the migratory activity of hepatocellular carcinoma cell lines. Int. J. Oncol. 2009, 35, 1409–1416 If available I would like to have more information about the structure of OPN secreted by the cardiomyocytes (lines 90 and 91)I found the parts 3 to 6 really interesting and well documented in human as well as in mouse and rat but difficult to read because of the information mix between the human studies and animal studies. I would suggest a better dichotomization between OPN function in human and in animals, to do not lose the important key messages at the end of the paragraphs and a stronger conclusion. As examples:
In the paragraph explaining OPN function in wound healing (lines 152 to 162), the information about human (reference 26) and mouse (reference 28) are mixed. In the paragraph describing OPN implication in cardiac fibrosis (lines 258 to 269), the human information (e.g. references 48 and 51) are not identified regarding mouse (e.g. references 49, 52 or 54) ore rat (reference 14) dataTo conclude, I appreciated the paper even if I was expecting a bit more details about the cardiac fibrosis according to the title and the difficulty to read it because of the uncertain organization and identification of the data.
Author Response
We thank the reviewer for their valuable feedback. Please find below our detailed response to the comments. Please note corrections have been included in both the edited and the final versions of the submitted manuscript.
In the part “osteopontin structure”, the OPN isoform notion is found line 109 when the explanation is in lines 117 to 121.
Response:
Thank you. We have re-examined part 2 Osteopontin structureto ensure that we are providing a coherent description of OPN’s structure and can summarize the following:
First mention and explanation of the OPN isoforms is in part 2 Osteopontin structurelines 166-167.In lines 167-170, we describe the structural differences between the different OPN isoforms.
In lines 170-177, we discuss the role of post translational modifications; which are critical to the function of OPN.
In lines 180, we discuss the OPN functional domains. In Figure 1 caption (line 322-328) we highlight the functional domains of OPN, which are conserved in all isoforms.
Furthermore, the novel involvement of OPN a, b and c isoforms in post-ischemic neovascularization and cell migration are found in part 4 where we highlight the role of OPN in the heart.
In the same way, some references appear in the wrong order: reference 59 before the references 57 and 58 or reference 63 before reference 62.
Response:
Thank you. We have rectified the order in which the references appear.
I would also be happy to see the relevant information from the references 61 and 65 which seems interesting for the manuscript but do not appear in the text.
Response:
Thank you. We have included the Gordin D et al., 2014 reference (previously reference 65, now 64), however we do not feel that the Hawkins NM et al., 2006 (previously reference 61) is longer relevant.
The introduction part is in my point of view well done despite an overusing of acronyms which are not evocated later in the text (for instance: I/R line 72, NMR line 112…)
Response:
We appreciate the Reviewer’s comment. We removed all unnecessary acronyms.
The “osteopontin structure” part presents some imprecise information:
Line 80: OPN was first described in malignant cells in 1979, in normal tissues in 1985 and renamed OPN in 1986. All references and details can be found here: Lamort AS, Giopanou I, Psallidas I, Stathopoulos GT. Osteopontin as a Link between Inflammation and Cancer: The Thorax in the Spotlight. Cells. 2019 Aug 2;8(8)
Response:
Thank you. We have included the information suggested and elaborated on the discovery of OPN.
The OPN with 314 aa (line 84) corresponds only to isoform OPN-a, the isoforms OPN-b and –c are respectively 300 and 287 aa. Some details can be found here: Chae, S.; Jun, H.-O.; Lee, E.G.; Yang, S.-J.; Lee, D.C.; Jung, J.K.; Park, K.C.; Yeom, Y.I.; Kim, K.-W. Osteopontin splice variants differentially modulate the migratory activity of hepatocellular carcinoma cell lines. Int. J. Oncol. 2009, 35, 1409–1416
Response:
Thank you. We have clarified this information in part 2 Osteopontin structure
If available I would like to have more information about the structure of OPN secreted by the cardiomyocytes (lines 90 and 91)
Response:
Thank you. To the best of our knowledge, no specific information about the structure of OPN by cardiomyocytes is available. As per the literature, the diverse biological functions of OPN can be attributable due to its structural features, post-translational modifications, interaction with multiple receptors, and different isoforms. As indicated, in the review article, three Opn cDNAs have been identified: Opn a (945 bp) encoding the full length protein; Opn b (903 bp cDNA) lacks a 14 amino acid sequence (amino acids 58–71; corresponding to 42 bp encoded by exon 4); and Opn c (864 bp cDNA) lacks an additional 27 amino acid sequence (amino acids 31–57). The functional significance of deleting this amino acid segment is unknown, although it is notable that it contains O-linked glycosylation and phosphorylation sites.
I found the parts 3 to 6 really interesting and well documented in human as well as in mouse and rat but difficult to read because of the information mix between the human studies and animal studies. I would suggest a better dichotomization between OPN function in human and in animals, to do not lose the important key messages at the end of the paragraphs and a stronger conclusion.
Response:
Thank you. We have re-arranged the evidence in animal models followed by clinical evidence from human tissue and patients. We have also included a summary Figure 3, which summarizes the role of OPN in cardiovascular diseases and cardiac fibrosis. We have also modified the conclusion as suggested.
To conclude, I appreciated the paper even if I was expecting a bit more details about the cardiac fibrosis according to the title and the difficulty to read it because of the uncertain organization and identification of the data.
Response: Thank you. We have revised the introduction paragraph to provide further details about cardiac fibrosis.

Round 2
Reviewer 2 Report
For this second reading of the review „Osteopontin: a Promising Therapeutic Target in Cardiac Fibrosis”, the authors enhanced the osteopontin (OPN) description as well as its implication in the diverse biological contexts. The focus on OPN in cardiac diseases was also improved and in general, I had a pleasure to read this second version of the review.
I also thank the authors for their detailed answers to my questions/comments. And I have just one question for this second round of review: I did not find any reference for figure 3 in the text. Would it be interesting to cite this figure in the conclusion as “summary” figure?